# Perceived Barriers to COVID-19 Testing

**DOI:** 10.3390/ijerph18052278

**Published:** 2021-02-25

**Authors:** Pearl A. McElfish, Rachel Purvis, Laura P. James, Don E. Willis, Jennifer A. Andersen

**Affiliations:** 1College of Medicine, University of Arkansas for Medical Sciences Northwest, Fayetteville, AR 72703, USA; RSPurvis@uams.edu (R.P.); DEWillis@uams.edu (D.E.W.); JAAndersen@uams.edu (J.A.A.); 2Department of Pediatrics, University of Arkansas for Medical Sciences, Little Rock, AR 72205, USA; JamesLauraP@uams.edu

**Keywords:** COVID-19 testing, research registry, barriers to testing, qualitative

## Abstract

(1) Background: Prior studies have documented that access to testing has not been equitable across all communities in the US, with less testing availability and lower testing rates documented in rural counties and lower income communities. However, there is limited understanding of the perceived barriers to coronavirus disease 2019 (COVID-19) testing. The purpose of this study was to document the perceived barriers to COVID-19 testing. (2) Methods: Arkansas residents were recruited using a volunteer research participant registry. Participants were asked an open-ended question regarding their perceived barriers to testing. A qualitative descriptive analytical approach was used. (3) Results: Overall, 1221 people responded to the open-ended question. The primary barriers to testing described by participants were confusion and uncertainty regarding testing guidelines and where to go for testing, lack of accessible testing locations, perceptions that the nasal swab method was too painful, and long wait times for testing results. (4) Conclusions: This study documents participant reported barriers to COVID-19 testing. Through the use of a qualitative descriptive method, participants were able to discuss their concerns in their own words. This work provides important insights that can help public health leaders and healthcare providers with understanding and mitigating barriers to COVID-19 testing.

## 1. Introduction

The Centers for Disease Control and Prevention (CDC) documented the first United States (US) laboratory confirmed case of coronavirus disease 2019 (COVID-19) on 22 January 2020 [1]. By February 2020, the US had over 26 million confirmed cases of COVID-19. The current diagnostic standard for COVID-19 diagnosis relies on real-time reverse transcriptase polymerase chain reaction (RT-PCR) for the detection of human severe acute respiratory syndrome coronavirus 2 (SARS-CoV-2) RNA in nasopharyngeal (NP) cells collected through a nasal swab [2,3]. In the early stages of the pandemic, widespread implementation of COVID-19 diagnostic testing in the US was hindered by the commercial availability of costly high-throughput instruments, the requirement of specific laboratory reagents and supplies, personal protective equipment (PPE), and geographic constraints that affected broad testing capacity [4,5]. Prior studies have documented that access to testing has not been equitable across all communities in the US, with less testing availability and lower testing rates documented in rural counties and lower income communities [6,7,8]. Communities with more minority residents have been shown to have lower availability of testing locations, resulting in lower rates of testing in minority communities. However, there is a limited understanding of the perceived barriers to COVID-19 testing. Therefore, the purpose of this study was to document the perceived barriers to COVID-19 testing.

## 2. Materials and Methods

### 2.1. Study Sample and Design

Participants were recruited from a volunteer research participant registry in Arkansas, ARresearch.org, established by the Translational Research Institute (TRI). ARresearch participants agreed to be contacted about research opportunities, including participation as healthy volunteers. The ARresearch registry was designed to be representative of the ethnic and racial diversity of Arkansas [9]. All potential participants (*n* = 4431) were identified for recruitment via email, inviting them to participate in a survey about COVID-19 testing. After removing 354 invalid or undeliverable email addresses, a total of 4077 recruitment email invitations were sent to valid email addresses. Recruitment emails provided a description of the study and gave potential participants the opportunity to document their consent. Before providing consent, potential participants verified that they met the study’s inclusion criteria. Inclusion criteria for the study required participants to be at least 18 years of age or older and living, working, or receiving health care in Arkansas. In order to eliminate duplicates, screening questions requested potential participants’ first and last names, dates of birth, and email addresses. Participants who completed the survey received a $20 gift card as remuneration.

REDCap (Research Electronic Data Capture) is a widely used web-based software designed for research data capture and management. REDCap was used to capture consent and survey responses for the study [10,11]. Survey questions related to participant demographics were sourced from the Behavioral Risk Factor Surveillance System (BRFSS) [12]. Open-ended questions were used to allow participants to provide short, written descriptions of any testing concerns they might have in their own words. Survey responses were collected during the months of July 2020 and August 2020. The study protocol was approved by the UAMS Institutional Review Board (IRB#261226).

### 2.2. Analytical Strategy

A qualitative descriptive analytical approach was used. A qualitative descriptive design summarizes participates’ experiences and the meanings participants ascribe to those experiences [13]. Two qualitative researchers reviewed responses and created an initial code book with emergent themes. All data segments were then coded and confirmation coding analysis was performed. Quotes that were most illustrative from the open ended survey questions were presented for each thematic domain. Given the more than 1200 participant responses, the number of quotes for each theme was vast, and only the most illustrative quotes are presented. The research team critically reviewed each analysis summary, ensuring analytic rigor and reliability by confirming that the data and illustrative excerpts were extracted to the correct thematic domain. Discrepancies in data interpretation were discussed twice per week over four weeks and resolved via consensus regarding the meaning of the quotes and where they best fit within the thematic coding. The goal of the descriptive qualitative method is to allow participants’ voices to be heard in their own words. In order to stay true to the participants’ responses, researchers left the punctuation and capitalization exactly as participants wrote them to allow participant’s written inflection to be reflected.

## 3. Results

Among the 4077 emails sent, 1288 individuals (31.6%) responded to the email invitation. Fifty-six participants were excluded because they did not meet the inclusion criteria for the study, and 11 participants were determined to be duplicates. When a record was determined to be a duplicate, the first response was selected to remain in the analytic sample while the second response was excluded. Therefore, from the 1288 who responded to the email invitation, 1221 participants were included in the sample. Descriptive statistics for participants’ age, race, education, insured status, and testing status are presented in Table 1. Most participants were between 35 and 64 years of age, women, college-educated, and reported having health insurance. A little more than two-thirds (69.5%) of the sample had not sought a COVID-19 test, and just under a quarter (22.7%) had been tested and received negative results.

The primary barriers to testing participants described were: (1) confusion and uncertainty regarding testing guidelines and where to go for testing; (2) lack of accessible testing locations; (3) perceptions that the nasal swab method was too painful; and (4) long wait times for testing results.

### 3.1. Confusion and Uncertainty Regarding Testing Guidelines and Where to Go for Testing: “Information about Testing Is Still Confusing”

Many participants expressed confusion about the testing guidelines and stated that “If I want to get tested, do I have to meet some minimum requirement? Do I have to pay? How much is it?” Even those participants who reported that they had searched for information said they were not able to find the clarity they desired. “I had difficulty Googling through a number of sites for highly disparate information about places/ways to get tested, how long it would take, cost, etc.” Another participant explained: “I did not know where to go for testing or how much it would cost. I looked online but information was scant, with few details on how, where or when tests were available. I drove to three different places before ultimately waiting a long time for a test.” 

To overcome this confusion, participants suggested that “there needs to be a single, publicly accessible repository of information in real time on testing sites, their address/phone/website, hours, eligibility, process, cost, types of tests.” Other participants echoed this recommendation and suggested a “clearinghouse that aggregated all that information at a searchable county level would be incredibly helpful.” Participants also suggested additional advertising about testing “locations listed via TV news stations and radio on a regular basis so we know where and when it's available,” and “I would like to see more information about when and how to get tested for COVID 19 circulated via local news, social media, etc.”

### 3.2. Lack of Accessible Testing Locations: “Please Make Testing More Accessible for All”

Participants stated that testing was not available to everyone in the community: “Testing is not assessable to the ordinary average person,” and others noted that “Drive-through clinics are inaccessible for low-income individuals without access to personal vehicles.” Participants voiced the need for testing locations that were more accessible by imbedding testing within local neighborhoods. Participants stated: “It would be nice to have localized testing,” and “I would like it if there was testing in individual neighborhoods.” Participants also discussed that testing in rural areas was limited or not available at all; “All testing is two or more hours drive away,” and “I do not believe there are enough testing sites in [participants’ rural] county.” Other participants noted that there had only “been one drive-thru testing conducted in my [participants’ rural county] but I didn't know until afterwards.” Common refrains in rural counties were “set up more test sites” and “build more test sites.”

### 3.3. Perceptions That the Nasal Swab Method Was Too Painful: “Something Less Painful Than Nose Swab”

Participants described their personal experience with the nasal swab as painful. Participants stated: “The test is very invasive and most individuals are refusing to take it because of the pain involved,” and “The testing is absolutely horrible. It would be nice to find a better way to test.” Many participants who had not previously been tested stated they had heard that the nasal swab COVID-19 test was too painful. Participants reported that the potential for pain deterred them from getting tested. Participants stated that they had heard it was “Extremely painful. I don’t want a stick crammed up my nose.” Other participants stated that they “have been told that it is quite painful. That has prevented me and several people I know from even getting tested. Isn’t there a less evasive and painful way to test?”

Participants who had not been tested described their fear of the nasal swab. “I am fearful of the pain associated with the nose swab;” and “painful and I am scared to go.” Participants said they “would drive farther to get tested another way that was not the nose swab.” Other participants stated they refused to get tested using nasal swabs. “We need to make the testing less painful. I refuse to get tested until they offer another method,” and “[I] refuse to be tested by nose swab. It’s too painful. Only testing I will do is a pain free option.” Overall, participants recommended “a more ‘patient-friendly’ testing method would be more persuasive in getting more people involved,” and that researchers “should come up with a new way of testing, because those who have been tested say its quite uncomfortable and irritating.”

### 3.4. Long Wait Times for Testing Results: “Testing Results Should Be Provided Faster”

Participants stated that one of the primary deterrents to testing was the time they had to wait for results. Using informal language, participants described their dismay with the delay in receiving their results: “Time frame for results? Why so long?” and, using more formal language, they stated: “I would have preferred to get results sooner. I had a COVID test on 8 July and received results on 13 July.” Several participants described two week delays in receiving their results. “My main concern is time for results. I would rather have an idea sooner instead of waiting up to 14 days,” and “it took 15 days to receive my results.” Participants stated that “Its important to have results quickly to minimize interruption and disruption of life.” Participants described how the delays in results were a burden on their lives: “took almost 2 weeks! Even though my entire family quarantined for that 2 weeks. Please get this fixed. I had family staying in a bedroom for 2 weeks away from other family members.” Participants described the financial burden that resulted from the delay in testing results: “testing is taking more than two weeks to receive them back. This is not good for employees who cannot return to work until test results received. People who mainly live paycheck to paycheck”.

Perhaps more concerning, participants described that the time delays created a perception that testing was futile. Participants explained: “Takes too long to get test results so there’s no point;” “took WAY too long to get results. Nearly two weeks. Was completely over by the time the results came;” and “It makes little sense to me to get a test that takes a week or two to get the results.” One participant succinctly articulated what many others had described: “takes too long to get test results so there's no point in trying.” Other participants described concerns that the delay in receiving results would increase the spread. “I would prefer rapid testing and results. Otherwise, it seems like the virus will continue to spread while people wait,” and “if the turnaround time for test results don’t decrease some, we could have infected people moving throughout our city for DAYS before they get their positive test results.” Participants voiced a desire for “a quicker turnaround time,” and “focus should be getting test results quicker, with others recommending that “the turnaround time for the results of the test to be NO more than 1–2 days.”

## 4. Discussion

This study sought to understand the barriers to testing using qualitative descriptive methods. Participants articulated significant confusion and uncertainty regarding testing guidelines and where to go for testing. CDC guidance as well as healthcare organizational policies, capacity, and practices have changed during the COVID-19 pandemic due to variations in supply and demand for testing [14,15,16]. These changes have left participants without a clear understanding of when and where they should go for testing. This is the first peer-reviewed article to document public confusion about guidelines for testing using qualitative methods. Public health and healthcare leaders can address this barrier by working together to create a collaborative online source for testing information in a community, taking care to systematically list the locations, hours of operation, testing criteria, and cost. 

Participants discussed the lack of accessible testing locations, especially in rural communities and communities with lower socioeconomic status. This finding is consistent with studies that have shown disparities in testing with low income, rural, and minority community members reporting being less likely to receive testing [6,17,18,19,20]. Participants went on to suggest that testing embedded within the community and leveraging neighborhood organizations and churches might be helpful. This finding is consistent with prior literature that shows that community-based organizations can be leveraged to reach community members for prevention activities and testing [21,22,23,24,25,26,27]. Public health and healthcare leaders can address this barrier by working together with community-based organizations to offer mobile testing in rural areas and in neighborhoods with fewer testing locations. For example, the states of Massachusetts, North Carolina, Texas, and West Virginia have increased access to testing by holding mobile testing units and events in non-traditional locations such as churches, schools, and community centers [8,21,22,23,24,25,26,27].

Participants described their perceptions that the nasopharyngeal swab method was too painful, echoing what has been reported by the media [28,29]. Even those who had not been tested were deterred from testing because of their fear of the pain associated with the nasopharyngeal swab. Among all sources of samples tested, those obtained from the lower respiratory tract and nasopharyngeal area are viewed to have the highest sensitivity, compared to saliva, sputum, blood, and feces [30]. In addition to the pain associated with health care worker-administered nasopharyngeal swabs, RT-PCR based assays are dependent upon sophisticated and labor-intensive centralized platforms, which are subject to delays in results reporting, as voiced by the participants of this study. Antigen tests, which detect specific viral proteins rather than viral RNA, are generally considered to have lower sensitivity than the RT-PCR tests. However, a recent study reported sensitivity rates that ranged from 89% to 98% for seven commercially available point-of-care antigen tests, using samples obtained during the first seven days of COVID-19 infection [31]. Antigen testing and point-of-care tests that can be self-administered may prove to be a more effective approach for diagnostic testing in the future [31,32].

Participants described long wait times for testing results, with many participants stating that they had to wait two weeks for results, which made testing seem futile. The finding is consistent with prior reports that reflect testing during the summer of 2020 when this study was conducted [33,34,35]. Although the increase in testing capacity nationally has improved (as of 6 November, Arkansas medical centers reported receiving a majority of test results within three days), the time between testing and return of results to patients is still at risk of significant delays as the number of COVID-19 cases rise [36]. This finding also highlights the importance of providing clear communication to patients about the need to quarantine as they wait for test results [37,38]. 

### Limitations

This study should be viewed in light of its limitations. Participants were recruited from a research registry in Arkansas. Within the study, 76.4% were White, 13.4% were Black and 6.7% were Hispanic. This is similar to the current census estimates which documents 72.0% of Arkansas’ population is White, 15.4% are Black, 7.7% are Hispanic. Although the sample size was diverse and large, the participants and their responses may not represent the general population. For example, some residents of Arkansas may not regularly use e-mail. Furthermore, 75% of the respondent were women, and this may influence the generalizability of the results. The large sample size ensured a sufficient sample of men, but a more balanced sample of men and women may have provided more nuanced insights. The written response data capture method did not allow for clarification or probing questions; however, the method did allow participants to communicate in their own words with a greater degree of confidentiality than individual or group interviews. Despite the limitations, this article makes a significant contribution to the literature as the first article to document barriers to COVID-19 testing among a large and diverse sample in Arkansas.

## 5. Conclusions

Disparities related to COVID-19 testing have been documented, however, little research has documented barriers leading to disparities in testing for COVID-19. Barriers to testing can perpetuate the spread of COVID-19. Thus, understanding these perceptions is critical to improving testing efforts. Through the use of a qualitative descriptive method, participants were able to voice their concerns in their own words. The perceptions articulated by participants suggest that some individuals may not obtain a COVID-19 test due to a variety of perceived barriers, including unclear information and confusion about where testing is available; a lack of accessible testing locations; concerns over the pain associated with nasal swabs; and the time it may take to receive results. The results presented here are the first to document participant-reported barriers to COVID-19 testing. For public health leaders and healthcare providers who seek to reduce the spread of COVID-19 and improve testing efforts, this work provides important insights that can help with understanding and mitigating barriers to COVID-19 testing.

## Figures and Tables

**Table 1 ijerph-18-02278-t001:** Descriptive statistics.

Variables	Number of Responses	Percentage
**Age Group** (*n* = 1221)		
18–34	316	25.9
35–64	712	58.3
65+	193	15.8
**Sex** (*n* = 1203)		
Women	905	75.2
Men	298	24.8
**Race and Ethnicity** (*n* = 1202)		
Black	161	13.4
White	918	76.4
Other	43	3.6
Hispanic (any race)	80	6.7
**Education** (*n* = 1202)		
High school or less	145	12.1
Some college	331	27.5
Four-year college degree	726	60.4
**Insurance Status** (*n* = 1002)		
Insured	950	94.8
Uninsured	52	5.2
**Coronavirus Disease 2019 (COVID-19) Nose Swab Test** (*n* = 1161)		
No, never tried	807	69.5
No, tried but unable	26	2.2
Yes, waiting on results	35	3.0
Yes, negative results	264	22.7
Yes, positive results	29	2.5

Note: Percentages do not include missing data. Percentages may not total 100 due to rounding.

## Data Availability

The data presented in this study may be made available upon request from the corresponding author. The data are not publicly available in accordance with funding requirements and participant privacy.

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
