# Peer review of "Perceived Barriers to COVID-19 Testing"

_ijerph, 2021, doi:10.3390/ijerph18052278_

Round 1
Reviewer 1 Report
This research would be useful during the covid pandemic, which is rapidly expanding, but a few weak points need to be corrected.
When studying via e-mail, it would be necessary to verify the person by phone call further. In case that the research by interview was impossible, using the formal questionnaire could decrease the ambiguity.
Major points.
1. The description after comparing the tested and untested people may be better to reduce confusion due to selection bias.
Author Response
This research would be useful during the covid pandemic, which is rapidly expanding, but a few weak points need to be corrected.
When studying via e-mail, it would be necessary to verify the person by phone call further. In case that the research by interview was impossible, using the formal questionnaire could decrease the ambiguity.
Response: We did not verify by phone call. Online survey research is used in many studies. We did verify that there were not duplications. See lines 60-62. “In order to eliminate duplicates, screening questions requested potential participants first and last name, date of birth, and email address.”
Major points.
1. The description after comparing the tested and untested people may be better to reduce confusion due to selection bias.
Response: We appreciate the reviewer’s comments, but this is beyond the scope of this study. The study sought to present the preferred location by sociodemographic rather than compare those who have and have not been tested.
Reviewer 2 Report
The authors conducted a survey study to understand the imitations of COVID-19 testing. The study has limitations that were mentioned by authors.
The study looks good on it's current form
McElfish et al conducted a survey study to identify barriers of COVID-19 testing.
How did authors identify study participants? is the 4,431 the total number of participants in the Arkansas registry?
What was the time frame the emails were sent and responses received?
The authors mentioned 46 patients were excluded? Why were these patients excluded?
I suggest for the authors to include the questions asked in a supplement material.
Why some responses are all in CAPS. I recommend not using caps for a whole sentence (for example, line 138, 139)
Were there any other responses that did not fit the categories mentioned
Author Response
Comments and Suggestions for Authors
The authors conducted a survey study to understand the imitations of COVID-19 testing. The study has limitations that were mentioned by authors.
The study looks good on it's current form.
McElfish et al conducted a survey study to identify barriers of COVID-19 testing.
How did authors identify study participants? Is the 4,431 the total number of participants in the Arkansas registry?
Response: This is the total in the registry that agreed to be contacted via e-mail. We have added that information. See line 53 and 54.
What was the time frame the emails were sent and responses received?
Response: Yes, this is the time that the e-mails were sent and received. See line 70.
The authors mentioned patients were excluded? Why were these patients excluded?
Response: We have added information about why participants were excluded. Fifty-six participants were excluded because they did not meet the inclusion criteria for the study, and eleven participants were determined to be duplicates. See line 89 and 90.
I suggest for the authors to include the questions asked in a supplement material.
Response: If requested by the editor we will include questions in the supplemental materials.
Why some responses are all in CAPS. I recommend not using caps for a whole sentence (for example, line 138, 139)
Response: The goal of the descriptive qualitative method is to allow participants’ voices to be heard in their own words. In order to stay true to the participants’ responses, researchers left the punctuation and capitalization exactly as participants wrote them to allow participant’s written inflection to be reflected. See lines 83-86. However, we are ok with removing the caps if the editors prefer to do so.
Were there any other responses that did not fit the categories mentioned?
Response: There were not any themes voiced by multiple participants that are not included.
Reviewer 3 Report
See attached document

Author Response
- The identification and selection of participants was based on the feasibility of them having an active email account. How was the representativeness of Arkansas' population sectors demonstrated? Especially with ethnic and racial diversity, which would involve a diversity of practices in the use of information technology. Although this was discussed and presented as a study limitation. 2. I recommend a brief explanation of how the sample represents the population, especially when only resident adults with work activity or receiving some health service were included in the study. How do they represent those who are not in these conditions?
Response: Within the sample, 76.4% were non-Hispanic White, 13.4% were non-Hispanic Black and 6.7% were Hispanic. This is similar to the current census estimates which show that of the approximately 3 million residents of Arkansas, 72.0% are non-Hispanic white, 15.4% are non-Hispanic black, 7.7% of residents report that they are of Hispanic (of any race). We have provided this information in line 241-243. We have also added more information to the limitations. See lines 245-248.
- What are the main advantages of the method used compared to other qualitative methodologies?
Response: The written response data capture method did not allow for clarification or probing questions; however, the method did allow participants to communicate in their own words with a greater degree of confidentiality than individual or group interviews. The written responses also allowed for rapid capture of more than 1200 responses. See lines 245-247.
- I suggest extending the description of the consensus used to reach interpretation agreements by the expert qualitative researcher’s team.
Response: We thank the reviewer for the comment and we have extended the description of consensus. See lines 83-84.
- In the description of results, it would be very worth reinforcing the analysis by indicating the percentage of consensus in the responses interpreted, as well as the number of participants that were grouped into each response category. Although the paper used qualitative methodology, this presentation would support the interpretation of the coauthors.
Response: We appreciate the recommendation, but we did not analyze results in a manner that quantize the information.
- The analysis and interpretation of the participants' responses could be discussed in terms of the public health strategy that has been implemented in Arkansas, so that perceived barriers could be compared with the construct of specific restrictions referred.
Response: While a full evaluation of the public health strategy that has been implemented in Arkansas is beyond the scope of this article, we have added information about the implications of this data to current and future public health strategy. See lines 189-192 and 209-212.
In addition, we have added funding. See lines 271-274. The project described was supported by the Translational Research Institute (TRI), grant UL1 TR003107 through the National Center for Advancing Translational Sciences of the National Institutes of Health (NIH). The content is solely the responsibility of the authors and does not necessarily represent the official views of the NIH.
Round 2
Reviewer 3 Report
Co-authors have improved the paper in terms of qualitative methodology.
Author Response
We thank you for your comment and feedback.